# SimTrack3D: A Simple Sequential Motion Modeling for Efficient 3D Single Object Tracking

## Abstract

Accurate tracking of objects in 3D point clouds requires continuous and efficient motion modeling across spatial and temporal dimensions. Although voxel-based methods have recently achieved strong performance thanks to rich BEV representations, they inevitably introduce redundancy, thereby complicating motion extraction. Building on efficient point representations and their sequential modeling, we venture that voxel features can be reformulated as a state-sequence paradigm, serving as an intermediate representation for more effective motion modeling. To this end, we introduce a serialized motion modeling framework that sequentializes BEV features by embedding spatial positions within a structured voxel grid, naturally enabling more efficient processing. At its core is a simultaneous spatiotemporal scanning mechanism that enforces causal inference, preserves structural priors, and jointly disentangles motion features across adjacent domains. By leveraging geometric priors with a compact yet precise representation, our framework leads to significantly reduced computational overhead while enhancing tracking performance. Interestingly, when integrated our approach with point-based methods, it further boosts performance through reinforced spatial modeling with minimal extra cost. Our method sets new SOTA records on the KITTI and NuScenes datasets, excelling in both accuracy and efficiency. Running at nearly 188 FPS on a single RTX 4090 GPU, it achieves a +19% speed improvement over the current best counterpart.

## 1 Introduction

With the rapid progress of LiDAR technology, 3D single object tracking (SOT) has attracted growing attention, serving as a cornerstone for many robotic functions such as navigation, manipulation, and autonomous driving. Unlike RGB or depth sensors, LiDAR measures the environment by emitting rays and capturing their reflections, producing sparse point clouds that directly encode the 3D geometry of the scene. This modality provides a natural basis for motion analysis by capturing the dynamics of objects across consecutive frames.

Early approaches relied on the directly accessible point representations (e.g., PointNet (Qi et al., 2017a;b)) and directly serialized local geometric positions into semantic features for downstream tracking (Qi et al., 2020). While computationally efficient, such methods lack explicit structural constraints, as points are processed independently, with spatial relationships only implicitly captured through pooling and merging (Fang et al., 2020; Xu et al., 2023a). As a result, the explicit positional encoding is often lost early in the aggregation process, and its capacity for robust motion modeling remains limited. To address this, recent methods have shifted toward voxelized BEV representations (e.g., VoxelNet (Zhou & Tuzel, 2018)), which expand point clouds into volumetric grids to explicitly encode spatial structure. Although more accurate, this comes at the cost of inflated dimensions and heavy computation, since processing large amounts of explicit spatial encoding becomes time-consuming and computationally demanding. Even with advances in optimized voxelization and processing (Zhou & Tuzel, 2018; Nie et al., 2025), these approaches still struggle to reconcile accuracy and efficiency, a critical trade-off for real-time applications.

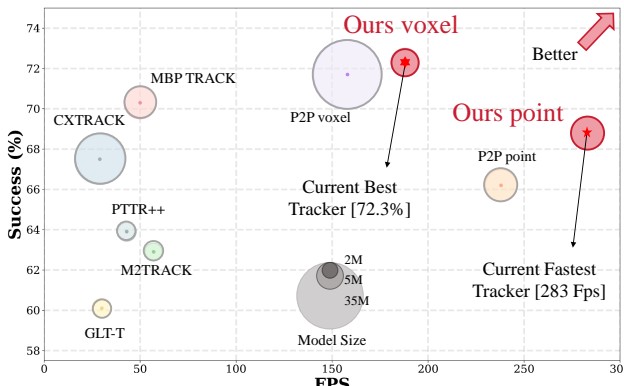

Figure 1: Comparison with state-of-the-art methods. We visualize mean success and Frames Per Second (FPS) performance across all categories on the KITTI dataset, where Ours-point and Ours-voxel denote the proposed tracking models with point-based representation and voxel-based representation, respectively.

Such an undesirable trade-off raises a fundamental question: is it possible to combine the efficiency of point serialization with the structural priors of voxelization? We venture that this is achievable by rethinking voxel features not as static grids, but as dynamic sequences. In fact, regardless of the point or voxel paradigm, both ultimately reduce 3D motion into a vector sequence, which is a natural observation, since the original point cloud itself is inherently a sequence of connected points. Therefore, such a reformulation should lead to a more compact yet structured representation that is naturally suited for motion modeling. In fact, motion in point clouds is inherently sequential—objects evolve smoothly across time, and their trajectories can often be captured more effectively by modeling structured temporal transitions rather than repeatedly aggregating redundant spatial encodings. By treating voxel features as state sequences, we can explicitly preserve spatial context while maintaining the efficiency and scalability of sequential processing.

To achieve this goal, we propose a paradigm that transforms voxelized BEV features into state sequences, allowing spatial structure to be embedded directly into sequential encoding. While recent advances in Structured State Space Models (SSMs) provide a promising framework for sequence modeling, their direct application to 3D SOT is nontrivial. Previous SSM-based approaches attempt to capture spatial context through scanning (Liu et al., 2024; Zhu et al., 2024b) and fixed-length spatial state modeling (Liang et al., 2024), but they mainly concentrate on single-frame local feature relationships and remain constrained for 3D SOT, since they cannot simultaneously balance layer-wise feature aggregation, encode cross-frame motion, and maintain computational efficiency.

These challenges motivate us to design a novel and tailored spatiotemporal sequence generation approach for 3D SOT. Specifically, we first extract target point clouds from historical bounding boxes and transform motion into dense BEV representations via voxelization and sparse convolution. We then employ four-directional spatial scanning of target BEV features to acquire shorter sequence features serving as templates. Subsequently, we obtain global BEV representations from the current point cloud scene and construct long sequence representations of scene point clouds as search sequences using identical four-directional scanning. By concatenating template sequences with search sequences, we derive cross-frame motion search sequences. Subsequently, through spatial state models, we perform variable-length aggregation of motion search sequences while preserving template features, efficiently integrating geometric spatial information into vector sequences through point serialization. Through this methodology, our tracker achieves state-of-the-art speed and performance.

Since point-based methods can be extended to voxel-based methods, we also analyze the feasibility of the reverse direction. Specifically, we introduce voxel geometric priors into shallow point-based networks through a simplified single-scan strategy. Remarkably, this design consistently improves performance while adding only minimal computational overhead, remaining within the same scale. This reveals that a more efficient sequence-based representation that goes beyond conventional representation-specific approache (Nie et al., 2025; Xu et al., 2023b; Zheng et al., 2022).

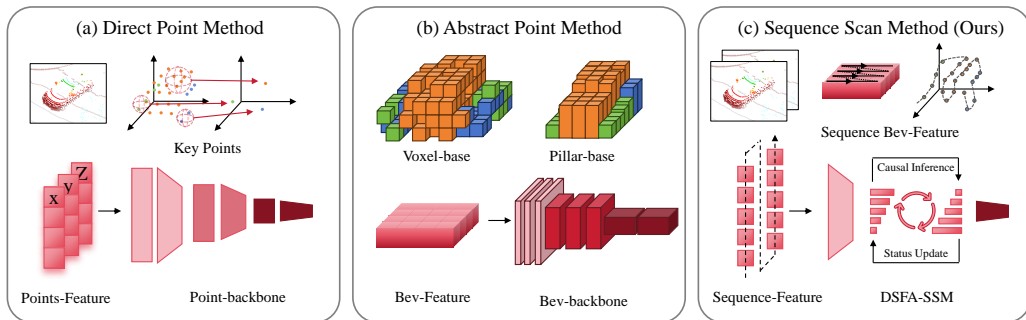

Figure 2: Three approaches for point cloud motion modeling framework: (a) direct point cloud processing; (b) abstract point cloud processing; and (c) our proposed sequential scanning processing.

In summary, the contributions of this work can be outlined as follows:

- We conducted in-depth analysis and exploration of the motion modeling structure, proposing a tracking framework named SimTrack3D that provides a novel perspective for efficient serialization of point cloud motion features.

- Our approach can be seamlessly applied to both point-based and voxel-based representations. By incorporating spatiotemporal motion modeling, it consistently boosts performance, particularly for voxel representations, while substantially reducing model parameters and inference time.

- We conduct comprehensive experiments on KITTI and NuScenes, demonstrating the leading performance and potential of our proposed framework.

## 2 RELATED WORK

### 2.1 STRUCTURED STATE SPACE MODELS

SSM (Gu et al., 2021b;a; Gupta et al., 2022; Li et al., 2022) is a class of RNN architectures tailored for sequence modeling. Recent advancements (Gu et al., 2021a; 2022; Nguyen et al., 2022; Gu & Dao, 2023) have elevated SSMs to achieve performance on par with Transformer-based models, while maintaining linear computational complexity. A particularly notable development is Mamba (Gu & Dao, 2023), which redefines conventional SSMs through input-dependent parameterization and hardware efficiency.

Building on the success of Mamba, ViM (Zhu et al., 2024a) and VMamba (Liu et al., 2024) adapt its 1D scanning mechanism to the 2D domain by introducing bidirectional and four-directional cross-scanning patterns, enabling the application of SSMs to visual data. A key methodology for integrating Mamba with visual priors is to design a visual-specific scanning pattern. Initial strategies include the bidirectional zigzag scan in ViM (Zhu et al., 2024a), the four-directional cross-scan in VMamba (Liu et al., 2024), and the serpentine patterns used in PlainMamba (Yang et al., 2024) and ZigMa (Hu et al., 2024), all of which preserve the spatial locality and multi-scale representation of image data. LocalMamba (Huang et al., 2024) and following works (Shi et al., 2024; Zhang et al., 2024; Li et al., 2024) inject spatial locality by segmenting the image into windows and performing localized scanning within each region. In contrast, the pioneer of SSM modeling for 3D point clouds is PointMamba (Liang et al., 2024), which uses this model to achieve global modeling with linear complexity through forward and reverse Hilbert structures. Despite the achievements mentioned above, in open and complex scenes, it is necessary to further design point cloud scanning strategies.

### 2.2 3D SINGLE OBJECT TRACKING

In the field of 3D single object tracking, the pioneering work is SC3D (Giancola et al., 2019). It uses a Kalman filter to generate a set of candidate 3D bounding boxes and selects the box with the

highest similarity to the given template target as the prediction result. However, SC3D is not an end-to-end framework and cannot run in real-time due to the excessive number of candidate boxes. The follow-up work P2B (Qi et al., 2020) proposes a 3D region proposal network leveraging VoteNet (Qi et al., 2019). It significantly improves tracking performance while achieving real-time speed. Based on this strong baseline, many follow-up works (Shan et al., 2021; Wang et al., 2021; Zheng et al., 2021; Zhou et al., 2022; Nie et al., 2023a;b; Xu et al., 2023a;b; Ma et al., 2023; Wu et al., 2025; Nie et al., 2024) have emerged. Taking recent methods as examples, CXTrack (Xu et al., 2023a) designs a target-centric Transformer network to explore contextual information for point-based motion. MBPTrack (Xu et al., 2023b) improves CXTrack through a memory network and box-prior localization network. M2Track introduces a motion paradigm for tracking by modeling the target point-based motion between consecutive frames to infer the target position. VoxelTrack (Lu et al., 2024) further explores fine-grained 3D spatial information for tracking by voxelizing unordered point clouds into 3D voxels and utilizing sparse convolution blocks for feature extraction, combined with a dual-stream encoder and cross-iterative feature fusion module. Noticing the scale relationship between tracking targets and scenes, P2P (Nie et al., 2025) simultaneously designs a part-to-part modeling network that fuses corresponding component information. Noticing the scale relationship between tracking targets and scenes, P2P simultaneously designs a network that fuses component motion. The designed P2P-point and P2P-voxel methods achieve optimal performance in speed and accuracy respectively. Although voxel-based motion modeling has achieved high performance, its model efficiency still needs improvement; point-based motion modeling benefits from direct point cloud sequence processing and has advantages in speed, but still falls short of explicit spatial encoding schemes in terms of spatial understanding. Considering that voxel-based methods still have significant redundancy in their BEV representations, our proposed SimTrack3D adopts spatiotemporal scan sequence motion features and implements variable-length Object-Scene template matching to achieve voxel-based motion modeling with fewer parameters.

## 3 Spatiotemporal Scanning and Dynamic Sequential Efficient Feature Aggregation

The spatiotemporal scan structure represents a simple modeling approach for serialized point cloud motion, which constructs point cloud motion priors that conform to SSM causal reasoning by re-ordering motion features. Sequential efficient dynamic modeling processes sequences through SSM structures in an autoregressive manner and aggregates motion features into sequential vectors. In the following sections, we first describe the fundamental principles of SSM and its 3D vision extensions. Subsequently, we introduce the core steps of Spatiotemporal Scan and Dynamic Sequential Efficient Feature Aggregation. In the final subsection, we explain the practical implementation and provide corresponding pseudocode.

### 3.1 Preliminaries

**State Space Models (SSM)** (Gu et al., 2021b;a; Gupta et al., 2022) originate from the analysis equations of linear time-invariant systems. Their main philosophy is to predict output responses through the superposition of input sequences and intermediate states. In implementation, they construct a transfer function that projects a one-dimensional input sequence $x(t) \in \mathbb{R}^L$ to an output response sequence $y(t) \in \mathbb{R}^L$ using a hidden state $h(t) \in \mathbb{R}^N$, where $L$ represents the sequence length and $N$ denotes the size of the hidden state. The entire sequence processing process can be represented by ordinary differential equations:

$$h'(t) = \boldsymbol{A}h(t-1) + \boldsymbol{B}x(t)$$
$$y(t) = \boldsymbol{C}h(t) \tag{1}$$

where matrix $\boldsymbol{A} \in \mathbb{R}^{N \times N}$ contains evolution parameters representing the compressed aggregation of historical hidden vector information, $\boldsymbol{B} \in \mathbb{R}^{N \times 1}$ and $\boldsymbol{C} \in \mathbb{R}^{1 \times N}$ are projection matrices that respectively represent the input's correction to the hidden vector and the hidden vector's contribution to the output. In the field of deep learning, this equation is first transformed into a discretized form through the zero-order hold rule (Gu et al., 2021b), treating the values of $x$ and matrices as constants over a small sampling interval $\Delta$ (Gupta et al., 2022), resulting in the discrete form:

$$h(t) = \overline{\boldsymbol{A}}h(t-1) + \overline{\boldsymbol{B}}x(t), \tag{2}$$

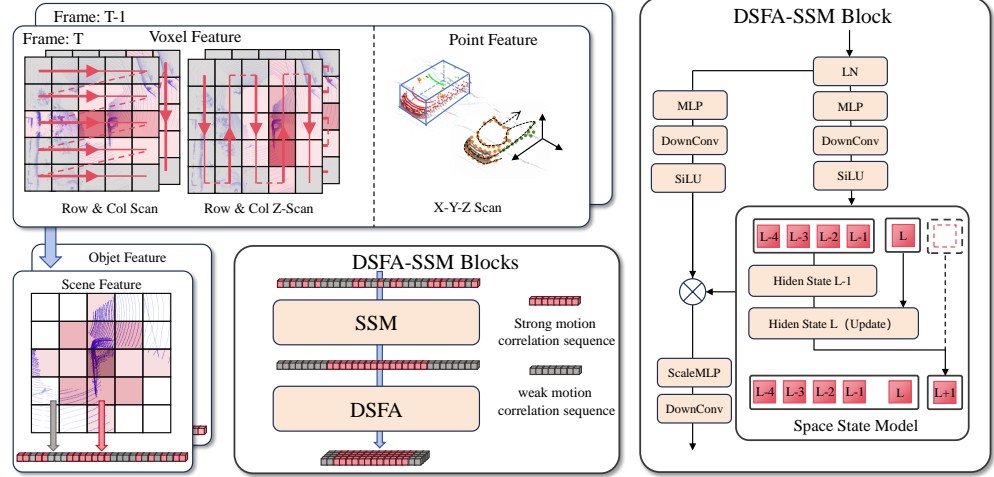

Figure 3: Spatiotemporal scanning DSFA-SSM modeling pipeline. For voxel-based methods, scanning is performed in BEV space, while for point-based methods, scanning is executed in x-y-z order. We concatenate the current frame's scene point cloud with the previous frame's target point cloud, and perform causal inference and feature aggregation of motion sequences through DSFA-SSM modules.

where $\overline{A} = \exp(\Delta A)$ and $\overline{B} = (\Delta A)^{-1}(\overline{A} - I) \cdot \Delta B$. The Mamba (Gu & Dao, 2023) scheme reformulates the corresponding matrices as input-dependent parameters, making $\overline{A}$, $\overline{B}$, and $\Delta$ learnable, and conducts hardware-aware parallel acceleration computation when solving the discretized equations.

**Vision SSM Models**. In the field of visual adaptation for SSM structures, pioneering works such as ViM (Zhu et al., 2024a) and VMamba (Liu et al., 2024) first employ zigzag scanning to transform 2D image features $\mathcal{F} \in \mathbb{R}^{H \times W \times C}$ into 1D sequences $\mathcal{F} \in \mathbb{R}^{N \times C}$, where $N$ is the number of pixel tokens, equal to the product of the height $H$ and width $W$ of the features. However, for 3D structures that do not conform to pixel arrangements, further abstraction of point clouds is required to achieve serialization of 3D features. The pioneering work in the field of 3D SSM architectures is Liang et al. (2024), which implements point cloud scanning through Farthest Point Sampling (FPS) and K-Nearest Neighbor (KNN) sampling of key points combined with bidirectional Hilbert space-filling curves to achieve linear SSM modeling. Its mathematical formulation can be expressed as:

$$
\begin{aligned}
S_l' &= \sigma\left(\text{DW}\left(\text{Linear}\left(\text{LN}\left(S_{l-1}\right)\right)\right)\right) \\
S_l'' &= \sigma\left(\text{Linear}\left(\text{LN}\left(S_{l-1}\right)\right)\right) \\
S_l &= \text{Linear}\left(\text{SelectiveSSM}\left(\boldsymbol{S}_l'\right) \odot \boldsymbol{S}_l''\right) + \boldsymbol{S}_{l-1}
\end{aligned}
\tag{3}
$$

$\boldsymbol{S}_l \in \mathbb{R}^{2n \times C}$ is the output of the $l$-th block, and $\sigma$ indicates SiLU activation (Hendrycks & Gimpel, 2016). The SelectiveSSM maps a state $x_t$ to $y_t$ through an implicit latent state $h_t \in \mathbb{R}^N$.

## 3.2 PIPELINE

As illustrated in the figure 3, spatiotemporal SSM modeling is primarily implemented through two key stages: (1) Causal matching between target templates and scene features, and (2) utilizing space-filling curves to serialize target motion sequences.

**Spatiotemporal Scan on Voxel and Point.** In the voxel-based approach, point clouds undergo voxel feature extraction via Voxel Feature Encoding (VFE) layers, subsequently yielding BEV feature maps. Serialized scanning feature vectors are obtained through predefined scanning encodings, where scanning paths are established directly during model initialization to minimize computational cost in individual forward inference passes. For the point-based approach, key point clouds are serialized through x-y-z ordered rearrangement. The computational procedures for both methods

can be formulated as follows:

$$\mathbf{S}_{voxel}^i = \text{VFE}(\mathbf{P_{raw}}, [B, C, H \cdot W]) \cdot \mathbf{P}_i^T$$
$$\mathbf{S}_{point}^i = \text{Reorder}(\mathbf{P_{key}}, [B, C, L]) \tag{4}$$

where $\mathbf{P}i \in \{0, 1\}^{(H \cdot W) \times (H \cdot W)}$ is the permutation matrix corresponding to the $i$-th scanning pattern, and $i \in \{\text{row\_major}, \text{z\_row}, \text{col\_major}, \text{z\_col}\}$ enumerates the four distinct spatial traversal strategies employed for feature reorganization. $\mathbf{P_{raw}}$ represents the original point cloud, whereas $\mathbf{P_{key}}$ refers to the downsampled key point set.

$$\mathcal{F}_{point}^t = \{\mathcal{S}_{xyz}\}^t$$
$$\mathcal{F}_{voxel}^t = \{\mathcal{S}_r, \mathcal{S}_{zr}, \mathcal{S}_c, \mathcal{S}_{zc}\}^t \tag{5}$$
$$\mathcal{F}_{motion}^t = \{\mathcal{F}_{object}^{t-1}, \mathcal{F}_{frame}^t\}$$

where $\mathcal{S}r$ represents the sequence scanned under the row-major principle, $\mathcal{S}zr$ denotes the zigzag row-major scanning sequence, $\mathcal{S}c$ represents the column-major scanning sequence, $\mathcal{S}zc$ denotes the zigzag column-major scanning sequence, $\mathcal{S}xyz$ represents the reordered sequence of key points following the x-y-z order, $\mathcal{F}_{frame}^t$ represents the features of the current scene, $\mathcal{F}_{object}^{t-1}$ denotes the features of target points from the previous frame, and $\mathcal{F}_{motion}^t$ corresponds to the motion modeling of the object. For the target motion sequences formed through the aforementioned scanning, the spatiotemporal structure of motion is encoded through the hidden state space $h(t)$ of SSM, thereby achieving explicit spatiotemporal modeling.

**Dynamic Sequential Feature Aggregation SSM (DSFA-SSM).** The motion sequences processed through spatiotemporal scanning undergo layer-by-layer feature aggregation, ultimately obtaining high-dimensional scene features for prediction, requiring gradual reduction of sequence scale and increase of channel dimensions. Through experiments, we found that good spatial modeling vectors can be obtained with fewer DSFA-SSM modules. This structure can significantly reduce model parameters and improve inference speed with fewer layers while using linear complexity to aggregate motion features. The structure of this part can be expressed as:

$$\mathcal{F}^i = \text{SSM}(\text{SiLU}^{(i)}(\text{DownConv}^{(i)}(\mathcal{F}_o^{i-1})))$$
$$\mathcal{F}_c^i = \text{SiLU}_c^{(i)}(\text{DownConv}_c^{(i)}(\mathcal{F}_o^{i-1})) \tag{6}$$
$$\mathcal{F}_o^i = \text{Scale\_Conv\_MLP}^{(i)}(\mathcal{F}c^i \otimes \mathcal{F}^i)$$

where $\mathcal{F}^i$ represents the features of the $i$-th layer, $\text{Scale\_Conv\_MLP}^{(i)}$ is used to compress the sequence length and increase the channel dimensionality, $\mathcal{F}_c^i$ denotes the dynamic control parameters for the $i$-th layer features, and $\mathcal{F}_o^i$ represents the output features of the current layer.

### 3.3 PRACTICAL IMPLEMENTATION

For spatiotemporal modeling of motion features, we adopt a predefined scanning encoding strategy. Specifically, the scanning patterns in voxel-based methods include: row-major scanning, Z-shaped row scanning, column-major scanning, and Z-shaped column scanning. The scanning patterns in point-based methods include: x-y-z permutation scanning, where each scanning pattern reorganizes high-dimensional motion representations into sequential features according to specific spatial orders. By causally concatenating the scanning results temporally, we obtain 1024-dimensional feature representations of inter-frame motion, providing diverse spatiotemporal perspectives for subsequent sequence modeling.

For sequential feature aggregation, we achieve enhanced feature extraction through reasonable scheduling of interactions between deep and shallow network layers. In shallow network layers, longer sequences are utilized to acquire more precise positional features, while in deep network layers, effective motion representations are extracted by progressively aggregating motion BEV sequence features and reducing the sequence scale. Consequently, we gradually aggregate the sequence scale from the maximum spatiotemporal features $\mathcal{F}^1 = 1024$ down to $\mathcal{F}^N = 1$.

In contrast, the sequence channel dimension ranges from $\mathcal{C}^1 = 256$ to $\mathcal{C}^N = 1024$, where $N$ corresponds to the depth of the feature aggregation backbone network, employed to balance training efficiency and model performance.

## 4 EXPERIMENT

### 4.1 EXPERIMENT SETTING

**Datasets.** We conduct experiments on point cloud SOT tasks. The point cloud training experiments adopt widely used public datasets, namely KITTI and NuScenes. For the division of training and test sets, we follow the experimental settings of P2P (Nie et al., 2025), dividing the 21 training sequences of KITTI into training set [0-17], validation set [17-19], and test set [19-21]. Compared to KITTI, NuScenes provides more challenging and large-scale scenes. NuScenes contains 700 training scenes and 150 test scenes.

**Evaluation Metrics.** Following mainstream evaluation practices, we employ One Pass Evaluation (OPE) (Wu et al., 2013) to assess tracking performance using Success and Precision metrics. Success calculates the intersection over union (IOU) between the predicted bounding box and the ground truth bounding box, while Precision evaluates the distance between the centers of the two corresponding bounding boxes.

---

**Algorithm 1** SimTrack3D with Spatiotemporal Scan DSFA-SSM

---

**Require:** motion_feature, scan_indices, DSFA-SSM_Blocks $\{\mathcal{U}_i\}_{i=1}^N$
**Ensure:** fused features $\mathcal{F}_{out} \in \mathbb{R}^{B \times L}$
 1: VoxelScan = {'row_major', 'z_row', 'col_major', 'z_col'}
 2: PointScan = {'x_major','y_major','z_major' }
 3: **for** each scan_type in scan_types **do**
 4:    indices = Scan(motion_feature, scan_type)
 5:    ThisFrame.append(feats_scanned)
 6: **end for**
 7: $\mathcal{F}^0$ = Concat(PrevFrame,ThisFrame)
 8: # DSFA-SSM Blocks Processing
 9: **for** $i = 1$ to $N$ **do**
10:    # SSM sequence $S^i$
11:    # Dynamic Control Sequence $C^i$
12:    $[S^i, C^i]$ =In_Projection$_{in}^i(\mathcal{F}^{i-1})$
13:    # SSM Autoregressive Inference
14:    **for** $j = 1$ to $L$ **do**
15:       # Update hidden state $H_{L-1}^i$
16:       $H_j^i = H_{j-1}^i$.Update($S^i$)
17:       $Sout_j^i = S_{j-1}^i$.Project($S^i$)+$H_j^i$.Evolution
18:       $Sout^i$.append($Sout_j^i$)
19:    **end for**
20:    # Feature Aggregation
21:    $\mathcal{F}_S^i$ = Scale_Conv_MLP$^i(Sout^i)$
22:    $\mathcal{F}^i = \mathcal{C}^i \otimes \mathcal{F}_S^i$
23: **end for**
24: # Final normalization and pooling
25: $\mathcal{F}_{out}$ = LayerNorm & Pool & Flatten($\mathcal{F}^N$)
26: Return: $\mathcal{F}_{out}$

---

### 4.2 COMPARISON
WITH STATE-OF-THE-ART TRACKERS

**Comparing model performance on the NuScenes dataset (Caesar et al., 2020).** We present a comprehensive comparison between our proposed methods and the previous state-of-the-art methods 1, including recent trackers on the NuScenes dataset, such as P2P (Nie et al., 2025) and MBP-Track (Xu et al., 2023b). Under the voxel-based processing paradigm, SimTrack3D exhibits superior performance, achieving the highest mean success rate of 60.20%. Under the point-based processing paradigm, SimTrack3D-point outperforms the previous leading method, namely MBPTrack (Xu et al., 2023b), by 1.68%. The framework further significantly improves tracking performance by effectively modeling the sequential features of motion components.

**Comparing model performance and efficiency on the KITTI (Geiger et al., 2012) dataset**.

To effectively evaluate the computational efficiency and generalization capability of our framework, we evaluated the model's performance and computational efficiency on the KITTI dataset. In comparison with P2P, which previously held state-of-the-art performance-efficiency trade-offs, our approach not only achieves optimal SUCCESS metrics but also outperforms P2P across both voxel-based and point-based models, thereby strongly demonstrating the inference speed superiority of the SSM architecture."

### 4.3 ABLATION STUDIES

**Research on Optimal Scanning Combinations and Causal Search.** To analyze our proposed sequential motion modeling, we conduct an ablation study on SimTrack3D-voxel on the NuScenes

Table 1: Comparisons with state-of-the-art methods on NuScenes dataset (Caesar et al., 2020).

| Track | Car [64,159] | Pedestrian [33,227] | Truck [13,587] | Trailer [3,352] | Bus [2,953] | Mean [117,278] |
|---|---|---|---|---|---|---|
| **SimTrack3D-voxel** | 65.64/72.82 | 46.10/75.15 | 65.74/66.50 | 68.60/64.80 | 54.00/50.42 | 60.20/72.12 |
| **SimTrack3D-point** | 66.40/73.60 | 41.02/68.11 | 64.74/66.32 | 68.55/66.32 | 56.25/53.46 | 59.16/70.75 |
| P2P-voxel (Nie et al., 2025) | 65.15/72.90 | 46.43/75.08 | 64.96/65.96 | 70.46/66.86 | 59.02/56.56 | 59.84/72.13 |
| P2P-point (Nie et al., 2025) | 62.14/68.45 | 39.68/65.59 | 62.50/63.44 | 69.04/65.14 | 57.90/55.46 | 55.92/66.64 |
| M²Track (Zheng et al., 2022) | 55.85/65.09 | 32.10/60.92 | 57.36/59.54 | 57.61/58.26 | 51.39/51.44 | 49.23/62.73 |
| MBPTrack (Xu et al., 2023b) | 62.47/70.41 | 45.32/74.03 | 62.18/63.31 | 65.14/61.33 | 55.41/51.76 | 57.48/69.88 |
| GLT-T (Nie et al., 2023b) | 48.52/54.29 | 31.74/56.49 | 52.74/51.43 | 57.60/52.01 | 44.55/40.69 | 44.42/54.33 |
| PTTR (Zhou et al., 2022) | 51.89/58.61 | 29.90/45.09 | 45.30/44.74 | 45.87/38.36 | 43.14/37.74 | 44.50/52.07 |
| BAT (Zheng et al., 2021) | 40.73/43.29 | 28.83/53.32 | 45.34/42.58 | 52.59/44.89 | 35.44/28.01 | 38.10/45.71 |
| PTT (Shan et al., 2021) | 41.22/45.26 | 19.33/32.03 | 50.23/48.56 | 51.70/46.50 | 39.40/36.70 | 36.33/41.72 |
| P2B (Qi et al., 2020) | 38.81/43.18 | 28.39/52.24 | 42.95/41.59 | 48.96/40.05 | 32.95/27.41 | 36.48/45.08 |
| SC3D (Giancola et al., 2019) | 22.31/21.93 | 11.29/12.65 | 30.67/27.73 | 35.28/28.12 | 29.35/24.08 | 20.70/20.20 |

Table 2: Comparisons with state-of-the-art methods on KITTI dataset (Geiger et al., 2012). Success/Precision are used for evaluation. We use "∗" to represent base tracking frameworks in the 3D SOT community.

| Tracker | Source | Car [6,424] | Pedestrian [6,088] | Van [1,248] | Cyclist [308] | Mean [14,068] | Hardware | Fps |
|---|---|---|---|---|---|---|---|---|
| **SimTrack3D-voxel** | Ours | 74.6/82.5 | 71.5/92.2 | 71.0/80.6 | 75.4/94.6 | 72.3/85.1 | RTX 4090 | 188 |
| **SimTrack3D-point** | Ours | 71.0/83.8 | 59.9/88.2 | 62.9/77.6 | 75.0/94.5 | 68.8/85.0 | RTX 4090 | 256 |
| P2P-voxel∗ (Nie et al., 2025) | IJCV'24 | 73.6/85.7 | 69.6/94.5 | 70.3/83.9 | 73.3/94.3 | 71.7/89.4 | RTX 4090 | 158 |
| P2P-point∗ (Nie et al., 2025) | IJCV'24 | 68.4/79.8 | 58.1/87.4 | 56.7/69.4 | 74.7/94.5 | 66.2/85.4 | RTX 4090 | 238 |
| M²Track++ (Zheng et al., 2023) | TPAMI'23 | 71.1/82.7 | 61.8/88.7 | 62.8/78.5 | 75.9/94.0 | 66.5/85.2 | Tesla V100 | 57 |
| M²Track (Zheng et al., 2022) | CVPR'22 | 65.5/80.8 | 61.5/88.2 | 53.8/70.7 | 73.2/93.5 | 62.9/83.4 | Tesla V100 | 57 |
| PTTR++ (Luo et al., 2024) | TPAMI'24 | 73.4/84.5 | 55.2/84.7 | 55.1/62.2 | 71.6/92.8 | 63.9/82.8 | Tesla V100 | 43 |
| MBPTrack (Xu et al., 2023b) | ICCV'23 | 73.4/84.8 | 68.6/93.9 | 61.3/72.7 | 76.7/94.3 | 70.3/87.9 | RTX 3090 | 50 |
| SyncTrack (Ma et al., 2023) | ICCV'23 | 73.3/85.0 | 54.7/80.5 | 60.3/70.0 | 73.1/93.8 | 64.1/81.9 | TITAN RTX | 45 |
| CXTrack (Xu et al., 2023a) | CVPR'23 | 69.1/81.6 | 67.0/91.5 | 60.0/71.8 | 74.2/94.3 | 67.5/85.3 | RTX 3090 | 29 |
| CorpNet (Wang et al., 2023) | CVPRw'23 | 73.6/84.1 | 55.6/82.4 | 58.7/66.5 | 74.3/94.2 | 64.5/82.0 | TITAN RTX | 36 |
| OSP2B (Nie et al., 2023a) | IJCAI'23 | 67.5/82.3 | 53.6/85.1 | 56.3/66.2 | 65.6/90.5 | 60.5/82.3 | GTX 1080Ti | 34 |
| GLT-T (Nie et al., 2023b) | AAAI'23 | 68.2/82.1 | 52.4/78.8 | 52.6/62.9 | 68.9/92.1 | 60.1/79.3 | GTX 1080Ti | 30 |
| CMT (Guo et al., 2022) | ECCV'22 | 70.5/81.9 | 49.1/75.5 | 54.1/64.1 | 55.1/82.4 | 59.4/77.6 | GTX 1080Ti | 32 |
| STNet (Hui et al., 2022) | ECCV'22 | 72.1/84.0 | 49.9/77.2 | 58.0/70.6 | 73.5/93.7 | 61.3/80.1 | TITAN RTX | 35 |
| PTTR (Zhou et al., 2022) | CVPR'22 | 65.2/77.4 | 50.9/81.6 | 52.5/61.8 | 65.1/90.5 | 57.9/78.2 | Tesla V100 | 50 |
| V2B (Hui et al., 2021) | NeurIPS'21 | 70.5/81.3 | 48.3/73.5 | 50.1/58.0 | 40.8/49.7 | 58.4/75.2 | TITAN RTX | 37 |
| BAT (Zheng et al., 2021) | ICCV'21 | 60.5/77.7 | 42.1/70.1 | 52.4/67.0 | 33.7/45.4 | 51.2/72.8 | RTX 2080 | 57 |
| MLVSNet (Wang et al., 2021) | ICCV'21 | 56.0/74.0 | 34.1/61.1 | 52.0/61.4 | 34.4/44.5 | 45.7/66.6 | GTX 1080Ti | 70 |
| P2B∗ (Qi et al., 2020) | CVPR'20 | 56.2/72.8 | 28.7/49.6 | 40.8/48.4 | 32.1/44.7 | 42.4/60.0 | GTX 1080Ti | 40 |
| SC3D∗ (Giancola et al., 2019) | CVPR'19 | 41.3/57.9 | 18.2/37.8 | 40.4/47.0 | 41.5/70.4 | 31.2/48.5 | GTX 1080Ti | 2 |

dataset. To investigate the impact of scanning strategies on model performance, we perform batch experiments on four typical row-by-row scanning structures proposed in this paper: $row\_major$, $col\_major$, $z\_row$, and $z\_col$. To maintain comparable scanning parameters, we use the same scanning sequence to repeatedly fill features. As shown in Tab. 3, when the scanning method lacks either horizontal or vertical information, the model will suffer from performance degradation due to the lack of spatial relationship information provided by the other scanning approach. Additionally, when single-category scanning methods exist, such as row and col, which are rotationally similar in spatial relationships, the limitation of sequential modeling in connecting spatially adjacent relationships will be highlighted, leading to decreases of 4.22 and 11.38 in Success and Precision, respectively. In contrast, partial scanning methods using non-similar structures, such as the combination of z_row and row, can preserve spatial information to some extent, with their performance degradation being lower than that of homogeneous scanning methods.

**On Causal template matching.** To validate the effectiveness of causal template matching, we utilize scene long sequences as templates to match target short sequences, employing identical four-directional BEV scanning. Given that initial point cloud features are lost during partial scanning modeling, this will lead to model performance degradation. Notably, although the sequence information of front and rear frames is reversed, which theoretically should significantly reduce the performance of sequential modeling, considering that during the scanning feature process, the model can learn the reverse representation of object

Table 3: Experimental results on NuScenes under various scanning configurations.

| Temporal scanning set | $z\_row$ | $row$ | $z\_col$ | $col$ | Success | Precision |
|---|---|---|---|---|---|---|
| Causal | ✓ | ✓ | ✓ | ✓ | 68.5822 | 64.7054 |
| | × | × | ✓ | ✓ | 66.7907 ↓1.79 | 61.1583 ↓3.55 |
| | ✓ | ✓ | × | × | 66.8414 ↓1.74 | 61.7378 ↓2.97 |
| | × | ✓ | × | ✓ | 64.3593 ↓4.22 | 53.3212 ↓11.38 |
| | ✓ | × | ✓ | × | 65.7138 ↓2.87 | 57.5992 ↓7.11 |
| Anti-causal | ✓ | ✓ | ✓ | ✓ | 66.4909↓-2.09 | 60.0828↓-4.62 |

box transformations. That is, although the model obtains reverse object motion during training, it can reverse this reverse object motion representation by fine-tuning a portion of parameters at the cost of sacrificing some model performance, to infer direct forward box motion.

**On Effectiveness of SSM.** To validate the effectiveness of our proposed DSFA-SSM structure for sequential feature aggregation, we tested various target categories on the nuScenes dataset while maintaining the same input-output structure for each DSFA-SSM layer and setting the number of blocks to 3. The results are shown in Figure 4. Due to the lack of effective sequential processing components, the model performance decreased. We found that for categories with scarce samples, the SSM structure performs worse than Conv1D, while for categories with larger data volumes such as cars, DSFA-SSM achieves leading performance. This demonstrates the dependence of the DFSA-SSM module on balanced training samples.

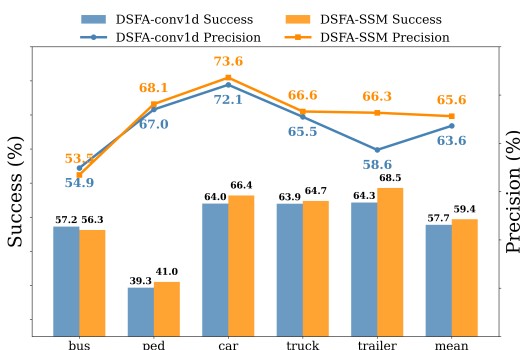

Figure 4: Model Performance Comparison between SSM and Conv Sequential Processing Methods.The SSM module demonstrates comprehensive advantages.

## 5 CONCLUSION

In this paper, we propose SimTrack3D, a tailored model structure for point cloud SOT tasks. Compared to mainstream convolutional and transformer approaches, SimTrack3D can more effectively model point cloud motion features by preserving the sequential nature of point cloud data structures. Given the neighborhood structure of point clouds and their voxel abstract data, along with the spatial locality of motion, SimTrack3D constructs motion sequences based on template matching through spatial traversal and temporal causal ordering. It employs carefully designed DSFA-SSM blocks to achieve long sequence spatial matching to short sequence semantic output with linear complexity, accelerating inference speed in an autoregressive manner while maintaining performance. Extensive experiments demonstrate that SimTrack3D's scanning approach is effective, achieving state-of-the-art experimental performance while enabling faster feature aggregation. However, more effective spatial scanning methods and modeling approaches that incorporate more historical motion data remain underexplored, which will be the focus of our future work.

## 6 REPRODUCIBILITY AND LLM USAGE STATEMENT

We have comprehensively ensured the reproducibility of this work, and all experimental results can be reproduced through the provided implementation and datasets. Specifically, we provide the following resources to promote reproducibility: (1) Code and Implementation: We will provide a reference framework of the SimTrack3D framework, including both point-based and voxel-based implementations, to be released as supplementary material. (2) Datasets and Processing: Our experiments are conducted on publicly available datasets (KITTI and NuScenes), with dataset splits and evaluation metrics consistent with the implementation described in the paper. (3) Experimen-

tal Setup: All hyperparameters, model architectures, and training configurations are clearly specified in configuration files. Detailed computational environment information, including PyTorch version (2.1.0+cu118), CUDA version (11.8), and hardware specifications (RTX 4090 GPU), has been documented to ensure consistent reproduction of results. (4) Performance Metrics: We report comprehensive and accurate performance metrics, including tracking accuracy, success rate, computational efficiency, and model parameter count. (5) Model Weights: Pre-trained model weights for SimTrack3D-point and SimTrack3D-voxel variants will be released after acceptance, enabling reproduction of our reported results without requiring retraining.

During the preparation of this manuscript and codebase development, we utilized Large Language Model (LLM) tools to assist with various aspects of the work. Specifically, LLM assistance was used for: (1) Writing and Language Improvement: LLM tools were used to enhance the clarity and grammatical accuracy of our technical writing, particularly for non-native English speakers. This includes assistance with sentence structure, terminology consistency, and overall manuscript readability. (2) Code Documentation and Comments: LLM assistance was used to generate comprehensive code documentation, inline comments, and README files to improve code readability and maintainability. (3) Technical Writing Support: LLM tools assisted in constructing technical explanations, ensuring consistent terminology usage, and improving the fluency of complex technical concept presentations in the paper.It is important to clarify that all core algorithmic contributions, experimental designs, mathematical formulations, and technical innovations presented in this work are original and independently developed. LLM assistance was limited to language improvement and documentation enhancement and did not influence the scientific content, methodology, or experimental results of our research. All experimental results, performance metrics, and technical contributions are entirely the product of our original research efforts, with LLM tools serving only as auxiliary aids for presentation and documentation purposes.

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

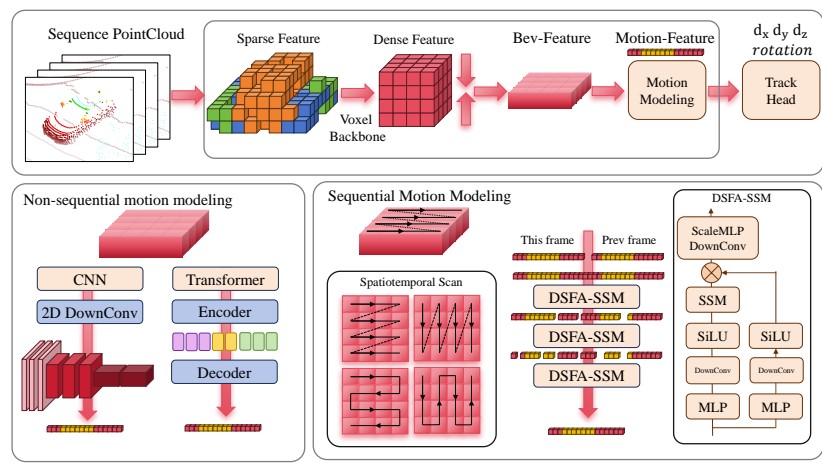

Figure 5: Algorithm Architecture of SimTrack3D-voxel.

| tracker | Model size | success | precision | FPS |
|---|---|---|---|---|
| SimTrack3D-point | 7.736839 | 68.8 | 85 | 283 |
| SimTrack3D-voxel | 5.621104 | 72.3 | 85.1 | 188 |
| CXTRACK | 18.3 | 67.5 | 85.3 | 29 |
| PTTR++ | 2.54359 | 63.9 | 82.8 | 43 |
| LTTR | 8.26 | 48.7 | 65.8 | 23 |
| M2TRACK | 2.25183 | 62.9 | 83.4 | 57 |
| GLT-T | 2.605417 | 60.1 | 79.3 | 30 |
| P2P point | 7.463178 | 66.2 | 85.4 | 238 |
| MBP TRACK | 7.38 | 70.3 | 87.9 | 50 |
| P2P voxel | 33.67359 | 71.7 | 89.4 | 158 |

Table 4: Partial Model Parameters Success Precision and FPS Metrics.

## A    APPENDIX

Our proposed SimTrack3D-voxel 5 point cloud processing pipeline first voxelizes the region of interest through Voxelize, then uses VFE to perform 3D sparse convolution on the scene, followed by dense operations on the sparse scene to obtain features in BEV. Motion modeling is employed to model motion, where mainstream processing pipelines include 2D feature aggregation and transformer-based token encoder-decoder methods. Our proposed method uses sequential modeling, preserves explicit spatial encoding, leverages SSM for efficient feature processing, and ultimately obtains high-dimensional semantic features of motion, which are input to the tracking head for predicting displacement and rotation parameters.

Reference data for Figure 1. Through serialized modeling, our method achieves optimal Success metrics and FPS metrics in point-based and voxel-based approaches respectively. Compared to P2P SimTrack3D, our proposed framework significantly reduces the parameter count for voxel-based methods and enhances the performance of point-based methods, thereby validating the effectiveness of our approach.

| kind | success | precision | point-mamba success | point-mamba precision |
|------|---------|-----------|---------------------|-----------------------|
| bus | 57.2181 | 54.8798 | 56.2530 | 53.4567 |
| ped | 39.2967 | 66.9931 | 41.0217 | 68.1129 |
| car | 63.9656 | 72.1122 | 66.3824 | 73.5940 |
| truck | 63.9440 | 65.4736 | 64.7419 | 66.5818 |
| trailer | 64.3027 | 58.6351 | 68.5494 | 66.3164 |

Table 5: Comparison of Success and Precision Metrics between DSFA-Conv and DSFA-SSM.

| kind | SimTrack3D(Deserialization) | SimTrack3D(serialization) | Pointnet(Deserialization) |
|------|----------------------------|---------------------------|---------------------------|
| car | 67.07/79.14 | 69.07/82.38 | 68.47/79.80 |
| cyc | 73.95/94.11 | 75.43/94.61 | 74.70/94.48 |
| ped | 57.22/83.90 | 61.75/87.60 | 58.13/87.42 |
| van | 46.70/55.62 | 57.11/69.70 | 56.71/69.41 |
| mean | 61.23/78.19 | 65.84/83.57 | 64.50/82.78 |

Table 6: Impact of Serialization on Model Performance Compared to Point-based Methods.

---

**Algorithm 2** Inference Pseudocode for SimTrack3D-point.

---

**Require:** prev_points, this_points, wlh_box_info
**Ensure:** predicted coordinates $\mathcal{C}_{out} \in \mathbb{R}^{B \times 3}$
 1: # Point Cloud Preprocessing
 2: Obj_pc = GetTargetPC(prev_points, wlh_box_info)
 3: (resequencing_Scene, resequencing_Obj) = ResequenceXYZ(Obj_pc, Scene_pc)
    // x-y-z Spatiotempora Scan
 4: stack_points = concat(resequencing_Obj, resequencing_scene) // Shape: [B, N, 3]
 5: # SimTrack3D Backbone Processing
 6: # Conv1D Feature Extraction
 7: $\mathcal{F}_{conv}$ = Conv1D(stack_points.transpose(1,2)) // 3→64
 8: $\mathcal{F}_{conv}$ = BatchNorm + ReLU($\mathcal{F}_{conv}$)
 9: #DSFA-SSM Blocks Processing
10: # DSFA-SSM Layer 1: 64→64
11: $\mathcal{F}^1$ = DSFA-SSM$^1$($\mathcal{F}_{conv}$) // scale=1
12: # DSFA-SSM Layer 2: 64→128
13: $\mathcal{F}^2$ = DSFA-SSM$^2$($\mathcal{F}^1$) // scale=2
14: # DSFA-SSM Layer 3: 128→1024
15: $\mathcal{F}^3$ = DSFA-SSM$^3$($\mathcal{F}^2$) // scale=8
16: # Global Feature Pooling
17: $\mathcal{F}_{global}$ = AdaptiveMaxPool1d($\mathcal{F}^3$) // [B, 128, 1]
18: # PointFuser Feature Fusion
19: prev_feats, this_feats = Split($\mathcal{F}_{global}$, dim=0)
20: **if** box_aware **then**
21:     wlh_feats = MLP(wlh_box_info) // 3→1024
22:     cat_feats = Concat(prev_feats, this_feats, wlh_feats)
23: **else**
24:     cat_feats = Concat(prev_feats, this_feats)
25: **end if**
26: # MLP Fusion Network
27: $\mathcal{F}_{fused}$ = MLPMixer$^1$(cat_feats) // 2/3→64
28: $\mathcal{F}_{fused}$ = BatchNorm + ReLU + Flatten($\mathcal{F}_{fused}$)
29: # PointHead_1024 Regression
30: $\mathcal{F}_{reg}$ = Linear + BatchNorm + ReLU($\mathcal{F}_{fused}$)
31: $\mathcal{C}_{out}$ = Linear$^4$($\mathcal{F}_{reg}$) // 128→6
32: **if** use_rotation **then**
33:     $\mathcal{R}_{out}$ = RotationHead($\mathcal{F}_{fused}$) // 1024→1
34:     Return: $\mathcal{C}_{out}[:, :3]$, $\mathcal{R}_{out}$
35: **else**
36:     Return: $\mathcal{C}_{out}[:, :3]$ // Only coordinates
37: **end if**

---

---

**Algorithm 3** Inference Pseudocode for SimTrack3D-voxel.

---

**Require:** prev_points, this_points, wlh_box_info
**Ensure:** predicted coordinates $\mathcal{C}_{out} \in \mathbb{R}^{B \times 3}$
1: # Point Cloud Preprocessing
2: Obj_pc = GetTargetPC(prev_points, wlh_box_info)
3: stack_points = concat(Obj_PC,Scene_PC) // Shape: [B, N, 3]
4: # Voxelization Process
5: # Dynamic Voxelization
6: voxels, coords = VoxelizationByGridShape(stack_points)
7: # Voxel Feature Encoding
8: voxel_features, coords = DynamicSimpleVFE(voxels, coords)
9: # Sparse 3D Convolution Backbone
10: # SparseEncoder Processing
11: $\mathcal{F}^1$ = SparseConv3D$^1$(voxel_features, coords) // 3→16
12: $\mathcal{F}^2$ = SparseConv3D$^2$($\mathcal{F}^1$, coords) // 16→32
13: $\mathcal{F}^3$ = SparseConv3D$^3$($\mathcal{F}^2$, coords) // 32→64
14: $\mathcal{F}^4$ = SparseConv3D$^4$($\mathcal{F}^3$, coords) // 64→128
15: # BEV Feature Generation
16: $\mathcal{F}_{bev}$ = SparseToDense($\mathcal{F}^4$, coords)
17: # BEV Feature Fusion with Spatiotemporal Scan
18: Obj_feats, Scene_feats = Split($\mathcal{F}_{bev}$, dim=0)
19: # Time Scan
20: cat_feats = Concat(Obj_feats, Scene_feats, dim=1) // [B, 256, 16, 16]
21: # Space Scan
22: scan_feats = MultiScanReorder(cat_feats) // row_major, z_row, col_major, z_col
23: merged_feats = Concat(scan_feats, dim=2)
24: # DSFA-SSM Processing
25: # Mamba-based State Space Model
26: $\mathcal{F}_{ssm}$ = MixerModel_SimTrack3D(merged_feats) // 3-layer Mamba
27: $\mathcal{F}_{global}$ = AdaptiveMaxPool1d($\mathcal{F}_{ssm}$) // [B, 1024, 1]
28: $\mathcal{F}_{global}$ = Flatten($\mathcal{F}_{global}$) // [B, 1024]
29: **if** box_aware **then**
30:     wlh_feats = MLP(wlh_box_info) // 3→1024
31:     $\mathcal{F}_{fused}$ = $\mathcal{F}_{global}$ + wlh_feats
32: **else**
33:     $\mathcal{F}_{fused}$ = $\mathcal{F}_{global}$
34: **end if**
35: # VoxelHead Regression
36: $\mathcal{F}_{reg}$ = Linear$^1$($\mathcal{F}_{fused}$) // 1024→512
37: $\mathcal{F}_{reg}$ = BatchNorm + ReLU($\mathcal{F}_{reg}$)
38: $\mathcal{F}_{reg}$ = Linear$^2$($\mathcal{F}_{reg}$) // 512→256
39: $\mathcal{F}_{reg}$ = BatchNorm + ReLU($\mathcal{F}_{reg}$)
40: $\mathcal{F}_{reg}$ = Linear$^3$($\mathcal{F}_{reg}$) // 256→128
41: $\mathcal{F}_{reg}$ = BatchNorm + ReLU($\mathcal{F}_{reg}$)
42: $\mathcal{C}_{out}$ = Linear$^4$($\mathcal{F}_{reg}$) // 128→6
43: **if** use_rotation **then**
44:     $\mathcal{R}_{out}$ = RotationHead($\mathcal{F}_{fused}$) // 1024→1
45:     Return: $\mathcal{C}_{out}[:,:3], \mathcal{R}_{out}$
46: **else**
47:     Return: $\mathcal{C}_{out}[:,:3]$ // Only coordinates
48: **end if**

---

