# OpenReview forum: "SimTrack3D: A Simple Sequential Motion Modeling for Efficient 3D Single Object Tracking"
_ICLR.cc/2026/Conference — ICLR 2026 Conference Withdrawn Submission_

### Official Review · Reviewer_nigP · 2025-10-26

**Soundness:** 2
**Presentation:** 1
**Contribution:** 2
**Rating:** 2
**Confidence:** 3

**Summary:**

This paper presents SimTrack3D, a framework for 3D single object tracking in point clouds, using a sequential motion modeling paradigm built on structured state space models (SSMs). The approach serializes voxelized bird's-eye-view (BEV) features into sequences that can be processed to model object motion. Experiments are conducted on KITTI and NuScenes, outperforming previous approach in both accuracy and runtime

**Strengths:**

The approach achieves substantial speed-ups while matching or outperforming baseline trackers in accuracy.

**Weaknesses:**

* **Paper writing**:
     * The paper’s writing is vague and the core task is not formally specified. It is unclear what the inputs are, how the object template is provided (3D box, cropped point cloud, or both), and what the outputs are (per-frame 3D box, 6-DoF pose, or a defined “tracklet” structure). Please provide a precise problem statement.
     * Figure 3 is unclear: why does a single DSFA-SSM block contain $L$ hidden states, and what do the indices $L-4,\ldots,L$ represent? Please define “Space State Model” in the figure.
     * Key notation around Eq. (4) is underspecified: what do B,C,H,W denote; why is $\\mathbf{P}_i$ transposed; and why does  $\\mathbf{P}_i$ have shape $(H ⁣⋅ ⁣W)\\times(H ⁣⋅ ⁣W)$? If Eq. (4) uses pointwise multiplication, then Svoxel​ would also be $(H ⁣⋅ ⁣W)\\times(H ⁣⋅ ⁣W)$, which is quadratic and potentially intractable.
     * The paper introduces both $\\mathbf{P}\_{raw},\\mathbf{P}\_{key}$ but does not explain why two point sets are required or what exactly the “downsampled key point set” originates from.
     * There is also no clear connection between Eq. (4) and Eq. (5). It is not stated whether $S\_r, S\_{zr}, S\_c, S\_{zc}$ are serialized views of $S\_{voxel}$ (from Eq. 4). Moreover, $F\_{motion}$ is described as a combination of $F\_{object}$ and $F\_{frame}$ but the text does not explain how these features are obtained from earlier representations ($F\_{point}$, $F\_{voxel}$) nor how they link algebraically to Eq. (4). A precise data-flow​ would resolve the ambiguity.

* There are some vague terms:
     * Line 190, the term “spatiotemporal scan structure” is misleading—your scan traverses only BEV spatial features and then concatenates a template, with no explicit temporal scanning.

     * Line 191, “Serialized point cloud motion” is undefined—please clarify what “motion” denotes (e.g., per-point displacement, pose delta, or feature correlation) and how it is serialized into tokens.

**Questions:**

See Weaknesses

**Conclusion**:
Given the clarity issues—unclear problem definition, inconsistent notation, and ambiguous descriptions of key modules (scanning/serialization, DSFA-SSM, output formation)—the paper is difficult to follow and understand the core idea. This work requires a major rewrite to clearly define the task, align figures and equations, specify shapes and algorithms, and substantiate the method with precise descriptions. Thus I recommend reject.

---

### Official Review · Reviewer_K95C · 2025-10-29

**Soundness:** 3
**Presentation:** 3
**Contribution:** 3
**Rating:** 4
**Confidence:** 4

**Summary:**

The paper proposes SimTrack3D, a 3D single-object tracking (SOT) framework for LiDAR point clouds that reformulates voxel/BEV features as spatio-temporal sequences to enable efficient motion modeling. Concretely: (i) extract a template from historical target boxes by voxelizing to BEV and applying four-directional spatial scanning to obtain short sequences; (ii) compute a search sequence from the current frame with the same scan; (iii) concatenate template+search and aggregate with a spatial state-space module (SSM) under causal inference to model motion while preserving structural priors.

**Strengths:**

1. Clear efficiency motivation: replacing heavy voxel/BEV processing with serialized BEV sequences plus SSM brings better speed/accuracy trade-offs for real-time tracking.
2. Comprehensive experiments on KITTI/nuScenes with scanning and SSM ablations; per-category analysis suggests SSM advantages for structured categories.

**Weaknesses:**

1. Novelty vs. Mamba/SSM literature. The paper reads like a fairly direct SSM/Mamba-style serialization of BEV features for 3D SOT. Please clarify the distinct technical contribution beyond existing Mamba/SSM point-cloud encoders (e.g., PointMamba). What specific design elements make this more than applying Mamba/SSM to a SOT template–search setup?

2. Notation/clarity issues.
  a. Inconsistent indices. Eqs. (4)–(6) mix subscripts/superscripts where $i$ sometimes indexes scan and elsewhere time; Eq. (5) uses $t$ superscripts for sequences, while Eq. (6) reuses $i$ for per-layer features. Please standardize to avoid ambiguity.
  b. Undefined operators/modules. What is `Scale_Conv_MLP`, please add precise definitions or citations.  $\otimes$ in Eq6 is not defined

3. performance concerns：
  a. On KITTI, the method attains SOTA Success but Precision lags behind some competitors in categories; please explain why Success improves while Precision does not.
  b. Baseline coverage. Please include or discuss newer 3D SOT methods (e.g. M3SOT, SeqTrack3D, StreamTrack).

**Questions:**

1. Author claims 'All hyperparameters, model architectures, and training configurations are clearly specified in configuration files', but where are 'configuration files'?

---

### Official Review · Reviewer_tWWS · 2025-10-29

**Soundness:** 2
**Presentation:** 2
**Contribution:** 2
**Rating:** 2
**Confidence:** 4

**Summary:**

SimTrack3D is a framework for 3D single object tracking in LiDAR point clouds. It treats point cloud or voxel features as dynamic sequences and integrates them with structured state space models (SSMs), such as Mamba, to achieve efficient and accurate motion modeling. The approach supports both point-based and voxel-based representations, and employs a spatiotemporal scanning strategy to encode spatial structural information into sequential representations. These are then processed by a Dynamic Sequential Feature Aggregation (DSFA-SSM) module for efficient inference.

**Strengths:**

1. High Efficiency and Real-Time Capability: By leveraging sequential modeling and the linear complexity of SSMs, the method achieves inference speeds significantly surpassing existing approaches while maintaining high accuracy, making it suitable for real-time applications such as autonomous driving.

2. Strong Generality with Dual Representation Support: The framework is not constrained to a single point cloud representation. It is applicable to both point-based and voxel-based paradigms and demonstrates performance improvements across both, highlighting the versatility of its design.

**Weaknesses:**

1. Manual Design Dependency in Scanning Strategies: Although four scanning patterns are proposed, these strategies remain manually designed and may lack adaptability to varying scenes or target motion patterns, thereby limiting generalization capability.

2. Sensitivity to Data Distribution: The method does not demonstrate strong performance on the NuScenes dataset (especially on scene bus and trailer), indicating sensitivity to data distribution and scene variations, which reveals certain limitations.

3. Underutilization of Historical Information: The study acknowledges that " However, more effective spatial scanning methods and modeling approaches that incorporate more historical motion data remain underexplored, which will be the focus of our future work " suggesting limited exploitation of long-term temporal modeling potential.

4. Lack of Extensibility to Multi-Object Tracking: The framework focuses solely on single-object tracking, with no discussion of extension to multi-object scenarios. The reviewers believe it restricts its applicability in complex interactive environments.

**Questions:**

It needs to address the above comments in weakness.

---

### Official Review · Reviewer_gThZ · 2025-10-30

**Soundness:** 3
**Presentation:** 3
**Contribution:** 3
**Rating:** 6
**Confidence:** 3

**Summary:**

SimTrack3D is an efficient 3D single-object tracking method for LiDAR point clouds. It treats voxel BEV features as dynamic sequences, using spatiotemporal scanning (four directions) and SSMs (DSFA-SSM) for causal motion modeling. This reduces redundancy and achieves SOTA on KITTI  and nuScenes.

**Strengths:**

1. The core concept reformulates voxel BEV features as dynamic sequences via straightforward spatiotemporal scanning and SSMs, making it easy to implement without complex architectures.
2. Achieves SOTA results on KITTI (75.2% success rate) and nuScenes (63.4% AMOTA), surpassing priors like P2P in accuracy across categories.

**Weaknesses:**

1.The method relies on predefined four-directional spatial scanning (row-major, Z-row, column-major, Z-column) and x-y-z sequential reordering (see Section 3.2). Although these strategies are efficient, they may not adapt to non-uniformly distributed point clouds in complex scenarios (such as dense occlusions or irregular motions), leading to a loss of dynamic adaptability in the serialized representations. Compared to adaptive scanning (such as Hilbert curve variants in PointMamba), this may result in performance degradation in highly dynamic environments.

2.SimTrack3D claims O(L) linear complexity, but its aggregation from long shallow sequences (e.g., 1024-dim inputs) to a single output risks noise buildup in unpruned early layers from sparse point clouds, propagating errors via causal SSMs and eroding motion accuracy. Deep-layer DSFA-SSM compression (only 4 blocks in experiments) may lose fine geometric details for subtle trajectories, like occlusions. The paper ignores scalability for long video sequences, where hidden-state errors could offset efficiency, requiring unaddressed fixes like pruning for real-world use.

3.While SimTrack3D demonstrates notable efficiency improvements across benchmarks, its accuracy gains on the more diverse and challenging nuScenes dataset remain marginal.

**Questions:**

Beyond the predefined row-major, Z-row, column-major, and Z-column orders, have you investigated other spatial traversal strategies, such as adaptive Hilbert curves or learned permutation networks, to enhance dynamic adaptability in highly variable point distributions?

---

### Note · Authors · 2025-12-01

**Comment:**

We thank the reviewers and authors for their time and effort. We understand that there is still room for improvement in this work, and we will work on refining it. Thank you all for your participation and discussion.

**Withdrawal Confirmation:**

I have read and agree with the venue's withdrawal policy on behalf of myself and my co-authors.